# Increasing uptake of FIT colorectal screening: protocol for the TEMPO randomised controlled trial testing a suggested deadline and a planning tool

Kathyrn A Robb ®,[1] Marie Kotzur ®,[1] Ben Young,[1] Colin McCowan,[2] Gareth J Hollands ®,[3] Audrey Irvine,[4] Sara Macdonald,[1] Alex McConnachie,[1] Ronan E O'Carroll,[5] Rory C O'Connor,[1] Robert J C Steele[6]

¹School of Health and Wellbeing, University of Glasgow, Glasgow, UK
²School of Medicine, University of St. Andrews, St Andrews, UK
³EPPI Centre, UCL Social Research Institute, University College London, London, UK
⁴Scottish Bowel Screening Centre, NHS Tayside, Dundee, UK
⁵Psychology, University of Stirling, Stirling, UK
⁶School of Medicine, University of Dundee, Dundee, UK

**Correspondence to**
Kathyrn A Robb;
Katie.Robb@glasgow.ac.uk

## ABSTRACT

**Introduction** Screening can reduce deaths from colorectal cancer (CRC). Despite high levels of public enthusiasm, participation rates in population CRC screening programmes internationally remain persistently below target levels. Simple behavioural interventions such as completion goals and planning tools may support participation among those inclined to be screened but who fail to act on their intentions. This study aims to evaluate the impact of: (a) a suggested deadline for return of the test; (b) a planning tool and (c) the combination of a deadline and planning tool on return of faecal immunochemical tests (FITs) for CRC screening.

**Methods and analysis** A randomised controlled trial of 40 000 adults invited to participate in the Scottish Bowel Screening Programme will assess the individual and combined impact of the interventions. Trial delivery will be integrated into the existing CRC screening process. The Scottish Bowel Screening Programme mails FITs to people aged 50–74 with brief instructions for completion and return. Participants will be randomised to one of eight groups: (1) no intervention; (2) suggested deadline (1 week); (3) suggested deadline (2 weeks); (4) suggested deadline (4 weeks); (5) planning tool; (6) planning tool plus suggested deadline (1 week); (7) planning tool plus suggested deadline (2 weeks); (8) planning tool plus suggested deadline (4 weeks). The primary outcome is return of the correctly completed FIT at 3 months. To understand the cognitive and behavioural mechanisms and to explore the acceptability of both interventions, we will survey (n=2000) and interview (n=40) a subgroup of trial participants.

**Ethics and dissemination** The study has been approved by the National Health Service South Central—Hampshire B Research Ethics Committee (ref. 19/SC/0369). The findings will be disseminated through conference presentations and publication in peer-reviewed journals. Participants can request a summary of the results.

**Trial registration number** clinicaltrials.gov NCT05408169

## STRENGTHS AND LIMITATIONS OF THIS STUDY

⇒ A suggested deadline for kit return and a faecal immunochemical test (FIT) planning tool are highly cost-effective (if effective) interventions that could be easily implemented in routine practice, through adding a sentence to the invitation letter and including a sheet of paper, respectively.

⇒ A 2x4 factorial randomised controlled trial (RCT) design will allow us to test the individual and combined impact of two behavioural interventions on colorectal screening uptake.

⇒ Integrating the RCT delivery into the existing colorectal screening process will provide evidence of the feasibility of providing either or both within a well-established national screening programme.

⇒ Cross-sectional and qualitative evaluation research will provide further insights into the psychological mechanisms that underlie any intervention effects.

⇒ While the study supports key priorities within the UK Cancer Strategies to increase uptake of FIT colorectal screening, if the interventions are effective, this will lead to an increase in demand on follow-up colonoscopy which currently has limited capacity.

## INTRODUCTION

Colorectal cancer (CRC) is the second leading cancer killer in Scotland and worldwide.[1 2] Screening has been shown to reduce deaths from CRC if enough people invited to participate.[3 4] The challenge is that high uptake is hard to achieve, and remains persistently below the target levels internationally. For example, screening rates are 44% in Australia and 62% in the USA.[5 6] Following the introduction of the single-sample faecal immunochemical test (FIT) in 2017, screening uptake in Scotland has risen to 65%, thus exceeding the Scottish target rate of 60%,[7] but remains just below the internationally accepted European Union target uptake rate of at least 66%.[8] Although this is higher than the European population-based CRC screening programmes average of 25%,[8] one-third of the population still do not screen

and uptake is unevenly distributed across socioeconomic factors leading to health inequalities.[7]

## Theoretical rationale

CRC screening uptake remains suboptimal, despite high levels of public enthusiasm. In a Scottish survey, 85% of people reported intending to complete the FIT,[9] yet participation is 65%.[7] This suggests that apparent enthusiasm for the new FIT is not being translated into uptake. A major reason for this disparity may be that people are 'not getting round to it',[10] reflecting the well-recognised intention-behaviour gap and consistent with the observation that much of our behaviour is determined by factors other than conscious, deliberative control.[11–14] This disparity is consistent with previous work on 'inclined abstainers',[13] showing that a proportion of people are inclined to participate in screening but fail to do it. This explanation seems particularly likely for CRC screening which, unusually for a screening test, is self-completed at home. General open-ended intentions to act often fail to lead to the desired behaviour.[15] Our previous research has found that people often put off or forget to complete and return the FIT.[16] Additional support through prompting the setting of FIT completion goals and planning may better support participation among those inclined to complete the FIT but who might otherwise fail to act.

## Goal setting

Goal setting, and particularly action-planning, can help initiate behaviour in those inclined to act. This is achieved by prompting a person to specifically plan when, where and how to act, which lessens the decision-making burden when that situation is encountered.[17–19] For example, a suggested deadline by which a task should be completed may prompt action planning. Currently, CRC screening invitations in Scotland provide detailed information about 'where' and 'how to act' but 'when to act' is left open, potentially reducing the likelihood that recipients will plan and act accordingly. In breast and cervical screening, providing a fixed appointment time increased uptake compared with an open invitation.[20] Similarly, in our recent qualitative study, women who attended breast and cervical screening but had not completed CRC screening described how the lack of an appointment facilitated the delay and forgetting of CRC screening: 'If it's a bowel screening one, yep, put it somewhere and think "Yes, I'll do that" and then forget about it because it doesn't have an appointment date. I think if something has an appointment date, you're forced to act' (54 years, not screened for CRC).[16 p6] Recent evidence suggests that both short (1 week) and long (3 week) deadlines achieve small but significant increases in FIT return.[21] Optimal deadline length, however, has yet to be determined. It is important to establish if a suggested deadline for FIT kit return can increase CRC screening participation compared with an open invitation in a population-based screening programme.

## Planning support

Systematic reviews clearly demonstrate that interventions supporting people to plan how to enact a behaviour (including action and coping planning) are particularly suited to changing the behaviour of inclined abstainers.[12] Indeed, planning support interventions have been shown to be more effective than other approaches, such as self-monitoring or providing instructions,[22] and have high adherence rates.[23] Planning interventions are often based on implementation intentions which promote behaviour change using 'if-then' plans and have successfully reduced alcohol consumption and smoking, and increased cancer screening attendance and physical activity.[11 12 24 25] Previous research on implementation intention interventions for CRC screening has had limited success when it has relied on prepared statements of barriers ('if') and possible solutions ('then').[26] Encouraging people to form their own implementation intentions for CRC screening appears to produce clinically significant increases in uptake.[24] Planning tools (also called volitional help sheets) emphasise this by requiring users to physically draw a line to link a barrier, for example, 'If I am tempted to eat when I am anxious' with a solution for example, 'then I will reward myself when I do not overeat' to ensure user-engagement.[27 p706] The rationale is that the 'if' barriers or critical situations are made personalised and salient and, as a result, the 'then' response or solution comes automatically to mind.

The Medical Research Council's recently updated guidance on complex intervention development and evaluation recommends addressing questions beyond intervention effectiveness, such as 'whether and how the intervention will be acceptable, implementable, cost effective, scalable, and transferable across contexts'.[28 p2] With a view to implementing a suggested deadline or planning tool at national scale, if effective, the acceptability of both interventions to CRC screening service users must be investigated.

## Objectives

The overall aim of the TimE fraMe and Planning tOol (TEMPO) study is to evaluate the impact of a suggested deadline and planning tool on FIT return in the Scottish Bowel Screening Programme.

The study addresses six research questions:

1. Does providing a suggested deadline increase uptake of CRC screening?
2. Does providing a FIT planning tool increase uptake of CRC screening?
3. Does providing both a suggested deadline and a FIT planning tool increase uptake of CRC screening?
4. Does the length of the suggested deadline (1, 2 or 4 weeks) impact on uptake of CRC screening?
5. What are the cognitive and behavioural mechanisms underlying any observed effect of providing a suggested deadline, a planning tool or a combination of both?

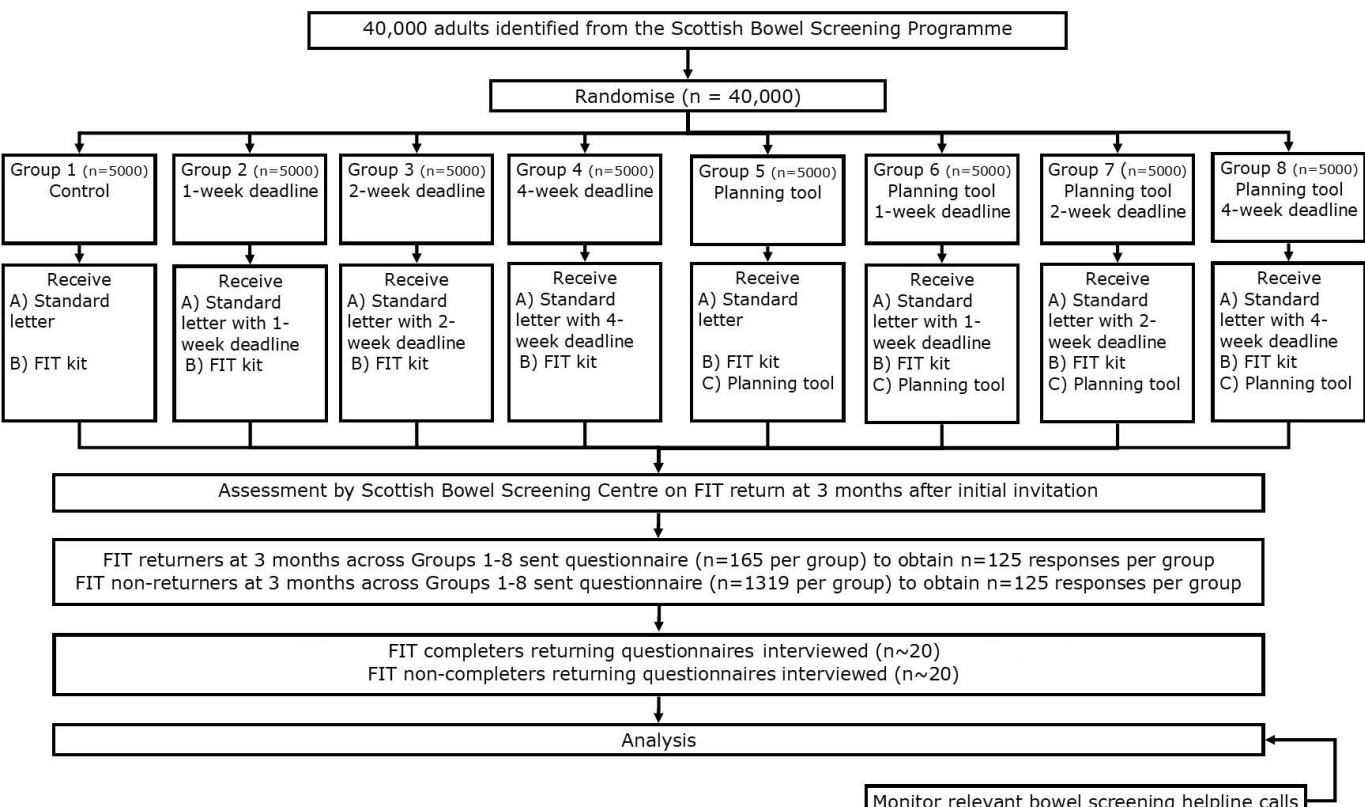

**Figure 1** CONSORT flow chart. CONSORT, Consolidated Standards of Reporting Trials; FIT, faecal immunochemical tests.

6. How acceptable do participants find receiving a suggested deadline, a planning tool or a combination of both?

## METHODS AND ANALYSIS
The study employs a 2×4 factorial randomised controlled trial (RCT) design combined with cross-sectional quantitative and qualitative evaluation research with a subsample of trial participants. Figure 1 shows the Consolidated Standards of Reporting Trials flow chart.

### Study setting
This is a single-centre trial to be conducted in collaboration with the Scottish Bowel Screening Centre based in Dundee, Scotland, and integrated into the national CRC screening programme. In Scotland, colorectal screening is offered to people aged 50–74, every 2 years, by single-sample FIT.

### Eligibility criteria
All people eligible for CRC screening in Scotland who have a Community Health Index Number can be included in this study.

### Recruitment and allocation
Following established recruitment procedures,[29 30] working in partnership with the Scottish Bowel Screening Centre staff, 40 000 consecutive adults due to be sent an FIT will be randomised to one of eight groups (figure 1) from 20 June 2022 over approximately 10 days.

The randomisation schedule will be computer-generated at the Robertson Centre for Biostatistics, University of Glasgow. The schedule will be stored in a secure area of the Robertson Centre for Biostatistics network, with no access for those developing the statistical analysis programmes and transferred via a secure web portal to the Scottish Bowel Screening Centre team, who identify when people are to be invited for CRC screening. The IT system electronically generates the appropriate mailing packages to be sent out to each person. For this study, randomisation will be performed by the method of randomised permuted blocks of length 8, so that in every eight packages sent, exactly one of each type will be included.

### Blinding
The process of generating and mailing the routine screening invitation letters is fully automated and is conducted by a large mail-handling company. A data file containing contact details for the screening invitation letters is sent to the mail-handling company daily. The company will print screening invitation letters with suggested deadlines and add the planning tool in accordance with the participants' allocation within this study. The researchers, including those performing the analysis, will therefore be blind to the allocation of participants. The participants will not be informed of the experiment.

**Table 1** Deadline intervention arms

| Intervention group | Suggested FIT return deadline | Deadline sentence |
|---|---|---|
| Groups 2 and 6 | 1 week deadline | **Please return your kit within 1 week (by [insert date 10 days from posting date]) or as soon as possible.** |
| Groups 3 and 7 | 2 week deadline | **Please return your kit within 2 weeks (by [insert date 17 days from posting date]) or as soon as possible.** |
| Groups 4 and 8 | 4 week deadline | **Please return your kit within 4 weeks (by [insert date 31 days from posting date]) or as soon as possible.** |

FIT, faecal immunochemical test.

## Interventions

There will be eight allocation groups (figure 1): group 1—standard invitation (n=5000); group 2—1 week deadline (n=5000); group 3—2 week deadline (n=5000); group 4—4 week deadline (n=5000); group 5—planning tool (n=5000); group 6—planning tool plus 1 week deadline (n=5000); group 7—planning tool plus 2 week deadline (n=5000) and group 8—planning tool plus 4 week deadline (n=5000).

### Deadline intervention

Those allocated to the deadline intervention (groups 2–4 and 6–8) will receive a screening invitation letter with one of three suggested deadlines added (table 1): 1 week, 2 weeks and 4 weeks. This choice was based on feedback from the Cancer Research UK Early Diagnosis Advisory Group, public and patient involvement representatives and observation of kit return patterns in the Scottish Bowel Screening Programme. The deadline will be displayed centrally on the standard CRC screening invitation letter as a bold, highlighted sentence (table 1). The wording *or as soon as possible* is intended to make clear that the deadline constitutes a suggested return date only, to avoid discouraging participants who are unable to meet the deadline. The existing invitation letter already states *post it as soon as possible* and the suggested deadline builds on this by providing a more explicit and specific goal. Furthermore, a standard reminder will be sent after 6 weeks without a deadline and so will provide a safety net for any potential participants who have missed the deadline in their first invitation letter.

### Planning tool intervention

Those allocated to the planning tool intervention (groups 5–8) will receive a planning tool with their screening invitation. The planning tool will be presented on a separate single sheet of paper inserted into the CRC screening invitation envelope. It asks participants to identify *concerns* (ie, barriers) they have with using the FIT from a list and to link them to a *tip* (ie, solution) from an adjacent list to help them overcome this concern. The planning tool reads, *Here are some tips that people find helpful to make the bowel screening kit even easier. Try drawing a line from any concern you have to a tip which might help you. You can draw as many or as few lines as you like. There are no right or wrong answers.* In conventional planning tool designs, concerns reflect critical situations and tips reflect solutions.[27] Each concern and tip is accompanied by an image illustrating its meaning. Finally, participants will have the option to write their own concern or tip. While the planning tool instructions encourage people to use the planning tool, they do not suggest that participants must do so in order to take part in CRC screening. The specific listed concerns and tips are based on extensive development work involving consideration of existing theoretical and empirical literature, in-depth qualitative interviews, a survey of people who had been invited to complete a FIT and codesign workshops to create a user-friendly planning tool with members of the public who have completed the FIT and others who have not completed the FIT.[31–33] The final planning tool was reviewed and approved by the Research Team, Patient and Public Involvement representatives (PPRs), and a patient and public group of the Scottish Primary Care Network.

Table 2 displays the concerns and tips shown in the planning tool. The planning tool is presented in online supplemental file 1.

### Control group

As the trial has a factorial design, it follows that the control group for the assessment of the independent effect of each intervention is all those who did not receive the intervention. This is because any impact of one intervention impacts to an equal extent those who do and do not receive the other intervention. Therefore, groups 1 and 5 together make up the control group for the assessment of suggested deadlines and groups 1–4 make up the control group for assessment of the planning tool. Only participants in group 1 will be sent a standard CRC screening invitation and FIT (without a suggested deadline or a planning tool) according to normal current practice. It is also important to note that three groups will receive both a planning tool and a suggested deadline, which will allow assessments of the combined effects of a suggested deadline and the planning tool compared with receiving a deadline only, a planning tool only or neither.

### Evaluation research of cognitive and behavioural mechanisms and acceptability

We will conduct two studies answering research questions 5 and 6.

**Table 2** Planning tool intervention—concerns and tips

| Concerns | Tips |
|---|---|
| If I feel scared about bowel screening… | … then I'll read the instructions. |
| If I keep putting off using the kit… | … then I'll think that this kit could help save my life. |
| If I am not used to using a kit like this… | … then I'll put the kit by the toilet to remind me. |
| If I am worried what it might find… | … then I'll wash my hands after using the kit. |
| If I think using the kit is messy… | … then I'll tell myself that I'm responsible for my health. |

## Questionnaire survey

We will conduct a questionnaire-based case–control study to assess the cognitive and behavioural mechanisms underlying any effect of providing a suggested deadline for FIT kit return and a planning tool.

### Recruitment

At least 3 months after the screening invitation, a subsample of trial participants will be identified from all eight groups, including people who returned (case) and did not return (control) their FIT, and people living in 40% most and least deprived neighbourhoods (based on Scottish Index of Multiple Deprivation (SIMD) quintiles).[34] The subsample will be sent a postal invitation to complete and return a survey. The aim is to obtain responses from approximately 1000 cases and 1000 controls across the eight groups. Based on previous work, we anticipate achieving a higher response rate from those returning the FIT and a lower response from those not returning the FIT, therefore approximately 11 872 questionnaires will be mailed, as shown in figure 1.[32] A reminder questionnaire will be sent to those who have not returned the questionnaire after 3 weeks.

### Materials and procedure

A brief questionnaire will include items to assess cognitive and behavioural mechanisms, for example, action and coping planning,[17] that were previously adapted for the CRC screening context.[19 32] It will also include a theory-informed acceptability measure that has been subject to a prevalidation method but does not yet have fully established psychometric properties,[35] as well as demographic characteristics. Participants will be asked if they are willing to participate in future research to assist recruitment for qualitative interviews.

## Qualitative interviews

We will conduct a qualitative interview study, adopting a phenomenological approach to explore participants' experiences of receiving an FIT with and without the interventions, and explore acceptability of the interventions.

### Recruitment

Interview participants will be sampled from survey participants who have indicated willingness to participate in further research. We will recruit approximately 40 participants and will aim to achieve a balance of participants who have and have not returned the FIT, men and women, from across the CRC screening invitation age range (50–74 years), from both deprived and affluent neighbourhoods (based on SIMD) and from the trial intervention groups. The estimated sample size is based on the principles of achieving data saturation and what is pragmatic, yet informative.

### Procedure

Potential participants will be contacted to request participation in a semistructured qualitative interview conducted face-to-face, by telephone or video-call. Interviews will last approximately 30–60 min, and topic guides (online supplemental file 2) will be used. Participants will be offered £30 towards travel and expenses.

## Outcomes

Our primary outcome is the proportion of FITs returned correctly completed to be tested by the CRC screening laboratory providing a positive or negative result, within 3 months of the FIT being sent to a person. The return and successful testing of any replacement FITs that were sent to people after the first FIT at study baseline will be treated as a FIT return. As secondary outcomes, we will evaluate cognitive and behavioural mechanisms and explore the acceptability of both interventions.

## Sample size

This 2×4 factorial RCT involves the assessment of two interventions, so each will be assessed at a significance level of 2.5%, to preserve an overall Type I error rate of 5%.

The assessment of the effect of the planning tool will involve the comparison of two groups of 20 000 participants. At a significance level of 2.5%, this gives 90% power to detect an increase in uptake rates from 65% to 66.8%. Alternatively, if the intervention increases uptake rates from 65% to 68%, the 97.5% CI for the difference in uptake rates will have a width of ±1.06%.

The assessment of the effect of suggested deadlines for FIT kit return will involve four groups of 10 000 participants. Comparing each suggested deadline with the control group (no deadline) at a significance level of 0.83% (one-third of 2.5%), there will be 90% power to detect increases in uptake rates from 65% to 67.63%. If an intervention increases uptake from 65% to 68%, the 99.17% CI for the difference in uptake rates will have a width of ±1.76%.

### Survey

The study is a case–control study (1000 cases, 1000 controls). The study will have 90% power at a 5% significance level to detect an OR for being a case of 1.146 per SD increase in a continuous predictor variable; for a

binary predictor with 50% prevalence, the study will have 90% power to detect an OR for being a case of 1.345; for a binary predictor with a prevalence of 10%, the study will have 90% power to detect an OR for being a case of 1.662.

### Trial data collection and management

For the purposes of this study, two additional variables will be added to the Scottish Bowel Screening Programme invitation data file: (1) a unique identifier for each participant; and (2) a variable coding for the allocated condition. The pseudonymised participant data including FIT kit return data will be transferred from the Scottish Bowel Screening Centre in Dundee to the Robertson Centre for Biostatistics at the University of Glasgow using a secure file transfer protocol.

### Data analysis plan

Primary and secondary outcome measures will be summarised using descriptive statistics.

#### Primary outcome

The primary analysis will use a logistic regression model, including a binary variable for whether the planning tool was provided, and a four-level categorical variable denoting which deadline was suggested. Age, sex and area deprivation (measured by the SIMD[34]) will be included in the model, to improve model specification. The main effects of each intervention will be reported as an OR, with a CI and p-value. For the estimate of the effect of the planning tool, a 97.5% CI will be reported. For estimates of the effect of alternative suggested deadlines, relative to no deadline, 99.17% CIs will be used. Subsequent analyses will model the interaction between the two interventions, to assess whether the interventions have independent effects or are synergistic. We shall also model interactions between each intervention effect and age, sex and deprivation, to assess whether the interventions have different effects for subgroups of the population. Assessing uptake by deprivation is important given the disparities in screening uptake by deprivation (53% most deprived vs 73% least deprived);[7] which any intervention could reduce or exacerbate.[36]

#### *Survey analysis*

Descriptive statistics, $\chi^2$ tests, analysis of variance and logistic regression will be used to examine cognitive and behavioural mechanism differences between the intervention and control groups and their association with FIT kit return.

#### *Qualitative analysis*

The interviews will be audio-recorded, transcribed, checked by the interviewer and analysed using the Framework Method.[37] This method facilitates systematic, rigorous and transparent data management, providing an audit trail from raw data to final themes. We will use NVivo software to undertake coding, manage and apply a working analytical framework, generate framework matrices and summarise and interpret data. This method

will adopt a combined deductive and inductive approach, guided by the behavioural mechanisms targeted by the interventions (eg, action and coping planning),[17] as well as constructs within the theoretical framework of acceptability,[38] while also allowing other experiences and any unintended intervention effects to be explored. BY will lead the analysis with support from KAR, MK and SM. Interpretation of findings may also involve the wider coinvestigator group, including the project PPRs, for improved credibility of findings. In addition, the results from the survey will be used to inform the analysis of the qualitative interviews and vice versa to determine whether synthesising the two data sources can provide unique insights which would not be apparent by analysing the data independently. Calls to the Scottish Bowel Screening Programme helpline relating to the suggested deadline for FIT kit return and planning tool will also be monitored via a call report form including date, time and nature of query as an additional assessment of acceptability.

### Participant timeline

This protocol describes a 3 month project as shown in figure 2.

### Study management and monitoring

This trial combines two individually funded studies with the same Principal Investigator. The trial will be managed by the coinvestigator teams of both studies including the Principal Investigator, academic coinvestigators, key researchers and PPRs who will oversee trial conduct and progress together. Data management, participant confidentiality and the conduct of all trial staff will adhere to the protocol (Version 1.4, 6 September 2021 or subsequent approved versions), Good Clinical Practice guidelines and the Data Protection Act 2018. An audit trail of all documentation and data collection will be kept in an electronic study site file to facilitate monitoring by the research team and the sponsor. This trial is sponsored by the NHS Greater Glasgow and Clyde. Amendments to the protocol will be notified to the sponsor and the responsible Research Ethics Committee for their approval. The sponsor may audit the trial at any time and without notice.

#### Stopping criteria

The CRC screening invitations and kits will be posted as one batch over approximately 10 days. There are no stopping criteria for recruitment.

#### Harms

This is a low-risk trial providing people eligible for the Scottish Bowel Screening Programme with additional written information. While we recognise that providing a suggested deadline or a planning tool may increase the burden of completing the FIT, we will ensure that participants: (a) are not disadvantaged if they cannot meet the suggested deadline for returning the FIT; and (b) are under no obligation to use the planning tool in order to complete CRC screening. We will be able to evaluate the

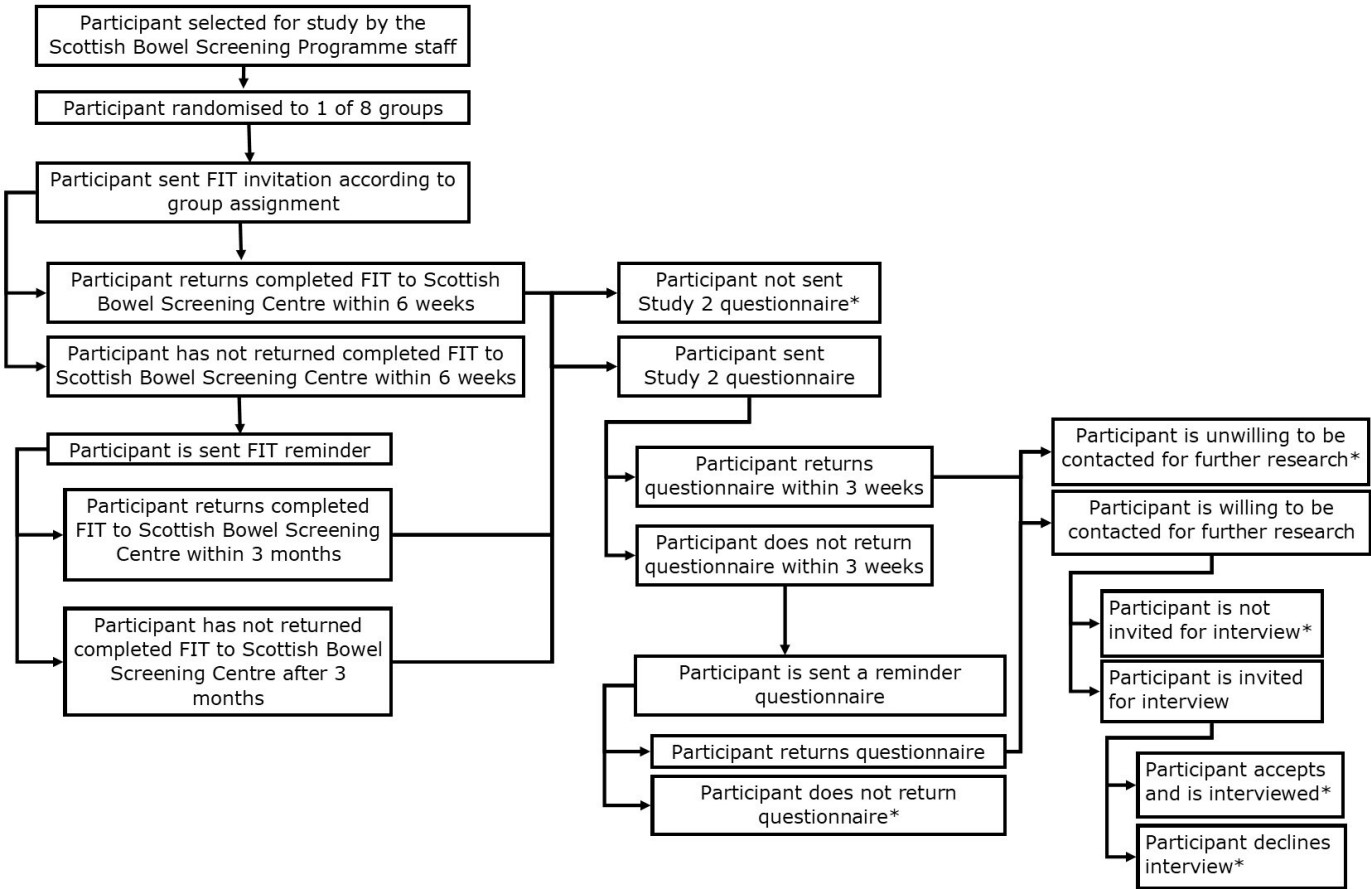

**Figure 2** Participant timeline note. *Data collection end points. FIT, faecal immunochemical test.

additional burden of each intervention in the questionnaire and interview evaluation studies.

## Public and patient involvement

The study is supported by two PPRs who have provided input throughout the programme of research. The PPRs were involved in the design of the study, the creation of materials to support the recruitment and join the coinvestigator meetings to discuss study progress. At the end of the study, the PPRs will review a plain language summary of the study findings which we will make available to the interview participants.

Intervention materials for this study, that is, the planning tool and the screening invitation letters including the three suggested deadlines, have been developed in workshops and focus groups and have received feedback from the PPRs, the Cancer Research UK Early Diagnosis Advisory Group and the PPRs of the Scottish Primary Care Research network.

## DISCUSSION

The purpose of this study is to increase uptake of FIT colorectal screening using two behavioural interventions. The study will evaluate the impact of a suggested deadline, a planning tool and the combination of a deadline and a planning tool on return of FITs. This research is important because CRC is the second most common

cause of cancer death globally[1] but uptake remains below target levels.

The study builds on existing theoretical and empirical evidence[11–17 27] and is the first attempt to test the individual and combined impact of a deadline and a codesigned planning tool within the real-world setting of a national screening programme. In addition, this study will provide evidence on the most effective timeframe for a suggested deadline as well as the cognitive and behavioural mechanisms that underlie any intervention effects. Finally, the study will investigate the acceptability of providing either intervention to those invited to participate in CRC screening. If effective, the interventions will be highly cost effective. A simple change to the wording of the colorectal screening programmes' invitation letters (deadline) and the inclusion of a sheet of paper (planning tool) will be all that is required to immediately see the benefits of increased uptake and subsequent earlier diagnosis of CRC.

The study has a number of potential limitations. While the study supports key priorities within the UK Cancer Strategies to increase uptake of FIT colorectal screening, if the interventions are effective, this will lead to an increase in demand on follow-up colonoscopy which currently has limited capacity. The primary outcome is return of a kit after 3 months rather than 6 months as used by the Scottish Bowel

Screening Programme. This was because the study was severely delayed by the COVID-19 pandemic and the follow-up was consequently reduced. It is unclear whether the results would be generalisable to screening programmes beyond the UK. The secondary outcomes examining the cognitive and behavioural mechanisms underlying any observed effects and the acceptability of the interventions will only be assessed after 3 months. Participants may not recall in detail their experience of receiving the deadline and/or the planning tool. Assessing the potential mechanisms immediately before and after test completion may have provided greater insight but this design risks contaminating the intervention itself. Our retrospective approach therefore provides the most appropriate option in this applied setting.

## ETHICS AND DISSEMINATION

Ethical approval was granted by the UK National Health Service South Central—Hampshire B Research Ethics Committee (REC ref. no. 19/SC/0369). Written informed consent will not be obtained before inclusion in the trial to maintain ecological validity and to avoid selection bias. This is a common approach in comparable trials performed in the context of national screening programmes,[29 30] without which it may not be viable to perform robust evaluations leading to evidence-based service improvements. There is very low risk of the interventions causing harm to participants and individuals will not be identifiable from trial data. Consent for the questionnaire survey will be implied through the completion of the questionnaire. Written informed consent to participate in a qualitative interview will be taken by the interviewer (online supplemental file 3). The research will be carried out in accordance with the World Medical Association Declaration of Helsinki (1964) and its revisions (Tokyo (1975), Venice (1983), Hong Kong (1989), South Africa (1996) and Edinburgh (2000)).

**Acknowledgements** We thank the patient and public representatives, Mary Cameron and Lucy J Robertson, for their continued support of this research, the patient and public involvement group of the Scottish Primary Care Research Network for their feedback on the intervention materials, Ann-Marie Digan of the Scottish Bowel Screening Programme, and Catherin McKay of ListenThinkDraw who provided illustrative services during the development of the planning tool and images for the final version.

**Contributors** KAR conceived both the planning tool and deadline interventions. KAR and MK designed the intervention procedures, study materials and evaluation with input from BY, SM, REO'C, RCO'C, AI, CM, AM, GJH and RJCS. AM conducted the sample size calculations and designed the statistical analyses. KAR submitted the original application for ethical approval and MK and BY drafted and managed approval amendments. KAR, BY and MK drafted the protocol manuscript. All coauthors read and approved the final manuscript.

**Funding** This study is supported by an Early Diagnosis Advisory Group (EDAG) Project Grant from Cancer Research UK (C9227/A27877—Increasing uptake of faecal immunochemical test (FIT) bowel screening: trial of providing a suggested deadline for FIT kit return, PI: KAR, coinvestigators: CM, AM, GJH and RJCS) and by a Response Mode Grant from the Scottish Chief Scientist Office (HIPS/17/23—Increasing uptake of bowel cancer screening: development of a FIT planning support tool, PI: KAR, coinvestigators: SM, REO'C, RCO'C, AI, RJCS and MK). The

funders have no involvement in the study design, data collection or analysis or the writing and publication of reports of the study. All researchers involved in this study were independent of the funder.

**Competing interests** None declared.

**Patient and public involvement** Patients and/or the public were involved in the design, or conduct, or reporting or dissemination plans of this research. Refer to the Methods section for further details.

**Patient consent for publication** Not required.

**Provenance and peer review** Not commissioned; externally peer reviewed.

**ORCID iDs**
Kathyrn A Robb http://orcid.org/0000-0002-1672-0411
Marie Kotzur http://orcid.org/0000-0001-6921-5075
Gareth J Hollands http://orcid.org/0000-0002-0492-3924

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
