## [Reviewer comments · BMJ Open]

ARTICLE DETAILS

TITLE (PROVISIONAL)	Increasing uptake of FIT colorectal screening: protocol for the TEMPO randomised controlled trial testing a suggested deadline and a planning tool
AUTHORS	Robb, Katie; Kotzur, Marie; Young, Ben; McCowan, Colin; Hollands, Gareth; Irvine, Audrey; MacDonald, Sara; McConnachie, Alex; O'carroll, Ronan; O'Connor, Rory; Steele, Robert

VERSION 1 – REVIEW

REVIEWER	de Jonge, Lucie Erasmus Medical Center, Public Health
REVIEW RETURNED	07-Nov-2022

GENERAL COMMENTS	This is an interesting study protocol about increasing FIT adherence in organized screening programs. As I am having experience conducted RCTs I will give comments only on study 1. The comments can be find below.  - Page 5, lines, are there any exclusion criteria after being randomized? What happens when individuals resign from the study? - Page 5, lines 32-34, were individuals at first randomized and they could opt-out in this study? Or do individuals need to send their consent? This is not clear. - Page 6, lines 22-23, was the invitation letter read by an independent reader? To be sure that all information is their in the appropriate language? - Page 6, lines 24-27, what will happen with individuals with a deadline? Will they end up in the control group after the reminder? Or will they be excluded from the study? - Page 7, lines 36-37, I am unsure about the authors choice of using group 1 and group 5 as the control group for the suggested deadline and group 1-4 for the planning toul as other than group 1, the groups are intervention groups and therefore the adherence response in those groups can be attributed to the intervention. If the authors would still choose this method, I would the authors to add some arguments for that. Why would it not have an effect on the outcomes (i.e. adherence)? - Is there room for any discussion or limitations, how to correct for certain biases? What the implication of this study could be?
--

REVIEWER	Goodwin, Belinda Cancer Council Queensland
REVIEW RETURNED	15-Nov-2022

GENERAL COMMENTS	This manuscript details the protocol for RCT designed to test the efficacy of two types of interventions to increase participation in the Scottish Bowel Cancer Screening Program: (i) a planning tool and
--

	(ii) a series of deadlines. The study is well-designed and the interventions have strong theoretical backing. I look forward to reading the results of the study! I have provided some minor suggestions below for clarity/ extra details. The only concern I had was that the SPIRIT checklist I downloaded was not complete. There were only page numbers listed for the top section. I assume these are all meant to be filled out. Perhaps an earlier version was uploaded? Happy to review this again if provided. Background Australia's screening rate is now 44% - still very low! But worth updating for accuracy. See https://www.aihw.gov.au/reports/cancer-screening/nbcsp-monitoring-2022/summary The background section might be strengthened by the addition of a sentence about why early detection is so important (e.g., statistics about lives-saved and/or success of treatment when caught in early stages) Methods Could the authors provide a short description of the Scottish Bowel Cancer Screening Program in the methods (age range; frequency; number of samples etc) for the international reader? The original letter has the words "post it as soon as possible" on it. Can the authors clarify where this text appears and in what context? For example, is it a general statement intended to encourage the participant to complete the kit and post it as soon as possible or is it relevant only to post instructions (i.e., intended as an instruction to post the kit as soon as possible after collecting the stool sample) I thought it was a little odd that the description of the methods second study was placed in the data analysis section. I would suggest moving this content earlier – not a big deal, but on my first read I wondered why this detail was missing. Could the authors provide a brief summary of the process of monitoring calls to the helpline. For example will calls be recorded (and participants consented) or will this involved consulting with phone operators/notes etc.? I note that authors do not specify hypotheses. I assume that it is predicted the interventions will increase participation. The authors could consider stating this or stating why they avoided setting hypotheses.
--	--

REVIEWER	Selva Olid, Anna Corporació Sanitaria Parc Taulí, Clinical Epidemiology and Cancer Screening
REVIEW RETURNED	23-Nov-2022

GENERAL COMMENTS	This is a well-written and well-structured protocol of project that contains three studies: A RCT (main study), a case control study and a qualitative study. I would like to provide some comments in order to improve the quality of the reporting and express some concerns regarding the sample size estimations.
---

	1. Bibliography. I would suggest updating some of the references. For example, there is a new publication of Globocan 2020. Also, references that support the effectiveness of CRC screening (ref 3 and 4) should ideally be randomized controlled clinical trials or systematic reviews of RCT. I would suggest adding some higher level evidence studies to support screening effectiveness. 2. I would suggest authors check if their reporting is in accordance with the SPIRIT statement (https://www.spirit-statement.org/) for reporting of protocols of RCT. I think the manuscript fulfill almost all the items and would suggest that authors state that they used this checklist in the reporting of their protocol. 3. I have concerns about an ethical aspect. Would participants be required to sign an informed consent? If the Ethical Committee stated that informed consent was not necessary for any reason in the RCT, it should be clearly stated. 4. Where substudies 2 and 3 also registered in a public platform? However, for substudies 2 and 3, I think an informed consent is needed. 5. I have doubts on how the sample size is calculated. Usually, sample size is established based on the minimum important difference that is expected to detect between groups in the main outcome (difference in participation rates), given an alpha and beta errors, and considering a proportion of dropouts. Authors just do the inverse thing: based on their sample, they calculate what is the minimum difference in participation that they would be able to detect at a determinate level of significance. I am not sure if this way of explaining sample size calculation is correct. Also, I think there is a mistake in line 3 of page 9: instead of 6.68%, should it be 66.85%? 6. Data analysis plan: Before conducting any multivariate model, a descriptive analysis and a bivariate analysis should be conducted. 7. For substudies 2 and 3, I would suggest to use the same structure that you have used for the methods section of the RCT: objective, study design, population, sample size, etc... 8. Regarding the substudy 2, please provide a definition on who will be the cases, who will be the controls and how will you select these samples. Would you apply any matching strategy between cases and controls? 9. I would like to suggest to authors to consider the use of electronic questionnaires and post-mail paper questionnaires and let participants choose the option they prefer. This can increase the response rate. 10. I have some concerns on how the power analysis of substudy 2 is conducted and reported. Sample size calculation is explained as if the main objective of the study was to find the OR of being a case or a control, and I think it makes no sense as authors select who will be cases and who will be controls. The objective of the study is to find differences in the prevalence of some "expositions" (cognitive and behavioural mechanisms) between cases and controls. 11. Please, explain is the questionnaires that will be used to measure acceptability are validated and in a population similar to yours and if they have good measurement properties. 12. Substudy 3. This is a qualitative study, but authors should explain which method are they going to follow phenomenology? Grounded theory?... 13. Regarding the sample size for the qualitative study, how do authors know that they would need 40 participants? Although they have estimated a sample size, sampling should continue until data saturation is reached (or it should be stopped is saturation is reached with less people). Recruitment, interviewing and analysis
--	--

	should be done at the same time to detect data saturation. 14. Please, provide a topic guide for the semi-structured interviews in an annex. 15. I would suggest authors to consider the conduction of focus groups (segregated by different profiles) instead of individual interviews. Individual interviews are an appropriate method for gathering data for their objective. However, focus groups allow interaction within participants and this interaction can let the emergence of new information. 16. Please, explain how many people will conduct the qualitative analysis. Would you use any software? 17. Authors should provide what methods they will use to guarantee the rigor of data: will how they will triangulate data?
--	--

REVIEWER	Khanna, Ajay Kumar Banaras Hindu University, Department of Surgery
REVIEW RETURNED	24-Dec-2022

GENERAL COMMENTS	The authors have planned the study protocol well and have addressed all the mentioned research questions. Following are my comments on the study protocol: 1. Sample-size: Page 9, line 3: Please check the typographical error “increase in uptake rates from 65% to 6.68%. Alternatively, if the intervention increases uptake rates from 60% to 68%, the 97.5% confidence interval for the difference in uptake rates will have a width of $\pm 1.06\%$.” 2. Will the participants in the intervention group receive some form of reward if they meet the deadlines? 3. As stated in the manuscript, no participant will be at a disadvantage if they don't meet the deadlines. How will the data for those participants who undertake screening after the deadlines will be part of the data analysis?
--

VERSION 1 – AUTHOR RESPONSE

Reviewer: 1

Dr. Lucie de Jonge, Erasmus Medical Center

This is an interesting study protocol about increasing FIT adherence in organized screening programs. As I am having experience conducted RCTs I will give comments only on study 1. The comments can be find below.

(1) Page 5, lines, are there any exclusion criteria after being randomized? What happens when individuals resign from the study?

Response: Thank you. There are no exclusion criteria after randomisation. Participants can not resign from the study as they will not be informed that they are in the study – see Response 2 to the Editor's comments.

(2) Page 5, lines 32-34, were individuals at first randomized and they could opt-out in this study? Or do individuals need to send their consent? This is not clear.

Response: Participants were not informed of the study and could not opt-out. Please see Response 2 to the Editor's comments.

(3) Page 6, lines 22-23, was the invitation letter read by an independent reader? To be sure that all information is their in the appropriate language?

Response: Yes. The letter was an amended version of the standard invitation letter used in the Scottish Bowel Screening Programme. As we stated in the 'Public and patient involvement' section: "Intervention materials for this study, i.e. the planning tool and the screening invitation letters including the three suggested deadlines, have been developed in workshops and focus groups and have

received feedback from the PPRs, the Cancer Research UK Early Diagnosis Advisory Group, and the PPRs of the Scottish Primary Care Research network.”

- (4) *Page 6, lines 24-27, what will happen with individuals with a deadline? Will they end up in the control group after the reminder? Or will they be excluded from the study?*

Response: Participants randomised to receive one of the three deadlines remain in the same group for the duration of the study including analysis.

- (5) *Page 7, lines 36-37, I am unsure about the authors choice of using group 1 and group 5 as the control group for the suggested deadline and group 1-4 for the planning tool as other than group 1, the groups are intervention groups and therefore the adherence response in those groups can be attributed to the intervention. If the authors would still choose this method, I would the authors to add some arguments for that. Why would it not have an effect on the outcomes (i.e. adherence)?*

- (6) Response: We thank Reviewer 1 for this suggestion but we do not agree. Since the study has a factorial design, then for the assessment of the effect of providing a deadline, the control group is all those who did not receive a deadline (Groups 1 and 5). It is true that half of those in this control group received a planning tool, but so did half of those who received a deadline. In that respect, if the planning tool has an effect on the likelihood of returning the FIT kit, it will impact those who receive a deadline and those who do not to an equal extent, and can be viewed as balanced between those randomised to receive a deadline or not. Similarly, whether a deadline was given, and what deadline was given, is balanced across those randomised to receive the planning tool or not, so for the assessment of the impact of the planning tool, groups 1 through 4 can be viewed as the control group. We have made a revision under the ‘Control group’ heading to emphasise that this approach is derived from the factorial design and briefly explain why this is appropriate. *Is there room for any discussion or limitations, how to correct for certain biases? What the implication of this study could be?*

Response: Thank you, this is a helpful suggestion. We have now added a brief Discussion including the strengths, limitations and potential implications.

Reviewer: 2

Dr. Belinda Goodwin, Cancer Council Queensland, University of Southern Queensland - Springfield Campus

- (1) *This manuscript details the protocol for RCT designed to test the efficacy of two types of interventions to increase participation in the Scottish Bowel Cancer Screening Program: (i) a planning tool and (ii) a series of deadlines. The study is well-designed and the interventions have strong theoretical backing. I look forward to reading the results of the study! I have provided some minor suggestions below for clarity/ extra details.*

The only concern I had was that the SPIRIT checklist I downloaded was not complete. There were only page numbers listed for the top section. I assume these are all meant to be filled out. Perhaps an earlier version was uploaded? Happy to review this again if provided.

Response: Thank you. The wrong version was uploaded in error. We have now included the completed version.

- (2) *Background Australia’s screening rate is now 44% - still very low! But worth updating for accuracy. See <https://www.aihw.gov.au/reports/cancer-screening/nbcsp-monitoring-2022/summary>*

Response: Thank you. We have now updated the manuscript with this more recent rate.

- (3) *The background section might be strengthened by the addition of a sentence about why early detection is so important (e.g., statistics about lives-saved and/or success of treatment when caught in early stages)*

Response: Our second sentence states, “Screening has been shown to reduce deaths from CRC...” We have updated our references in response to your comment and Reviewer 3, comment 1.

- (4) *Methods Could the authors provide a short description of the Scottish Bowel Cancer Screening Program in the methods (age range; frequency; number of samples etc) for the international reader?*

Response: Thank you for this helpful suggestion. We have now added the following information to the Study setting: “In Scotland, colorectal screening is offered to people aged 50-74, every two years, by single-sample faecal immunochemical test (FIT).”

- (5) *The original letter has the words “post it as soon as possible” on it. Can the authors clarify where this text appears and in what context? For example, is it a general statement intended to encourage the participant to complete the kit and post it as soon as possible or is it relevant only to post instructions (i.e., intended as an instruction to post the kit as soon as possible after collecting the stool sample)*

Response: The text appears on the back of the invitation letter which focuses on instructions for completing the kit. The full sentence reads, “Put the finished test in the pre-paid envelope and post it as soon as possible.” Please see: https://www.healthscotland.com/uploads/documents/30055-__Bowel%20screening%20letter-August2021-English.pdf

- (6) *I thought it was a little odd that the description of the methods second study was placed in the data analysis section. I would suggest moving this content earlier – not a big deal, but on my first read I wondered why this detail was missing.*

Response: We agree and have moved it earlier.

- (7) *Could the authors provide a brief summary of the process of monitoring calls to the helpline. For example will calls be recorded (and participants consented) or will this involved consulting with phone operators/notes etc.?*

Response: Helpline staff were provided with a Frequently Asked Questions document to support response to calls concerning the study. Helpline staff were also asked to complete a report form ‘tick list’ which recorded the date, time and nature of the queries they received regarding the study. We have added this detail (underlined here) to the methods: “Calls to the Scottish Bowel Screening Programme helpline relating to the suggested deadline for FIT kit return and planning tool will also be monitored via a call report form including date, time and nature of query as an additional assessment of acceptability.”

- (8) *I note that authors do not specify hypotheses. I assume that it is predicted the interventions will increase participation. The authors could consider stating this or stating why they avoided setting hypotheses.*

Response: Thank-you for your comment; we went with research questions rather than hypotheses specifically. However, in research questions 1 to 3 we specify that we are investigating whether the interventions increase uptake and in question 4 we talk about impact on screening.

Reviewer: 3

Dr. Anna Selva Olid, Corporació Sanitaria Parc Taulí

This is a well-written and well-structured protocol of project that contains three studies: A RCT (main study), a case control study and a qualitative study. I would like to provide some comments in order to improve the quality of the reporting and express some concerns regarding the sample size estimations.

- (1) *Bibliography. I would suggest updating some of the references. For example, there is a new publication of Globocan 2020. Also, references that support the effectiveness of CRC screening (ref 3 and 4) should ideally be randomized controlled clinical trials or systematic reviews of RCT. I would suggest adding some higher level evidence studies to support screening effectiveness.*

Response: Thank you for this suggestion. We have updated references 1 and 2 to the more recent data. We have also updated the references for the effectiveness of CRC screening [3,4].

- (2) *I would suggest authors check if their reporting is in accordance with the SPIRIT statement (<https://www.spirit-statement.org/>) for reporting of protocols of RCT. I think the manuscript fulfill almost all the items and would suggest that authors state that they used this checklist in the reporting of their protocol.*

Response: Thank you. The wrong version was uploaded in error. We have now included the completed version.

- (3) *I have concerns about an ethical aspect. Would participants be required to sign an informed consent? If the Ethical Committee stated that informed consent was not necessary for any reason in the RCT, it should be clearly stated.*

Response: Thank you. Please see our response to the Editor’s comment 2 which addresses this concern. In addition, and to increase clarity, we have moved the ethics section earlier in the manuscript and have added the sentences, “Written informed consent will not be obtained before inclusion in the trial to maintain ecological validity and to avoid selection bias. This is a common

approach in comparable trials performed in the context of national screening programmes, without which it may not be viable to perform robust evaluations leading to evidence-based service improvements. There is very low risk of the interventions causing harm to participants and individuals will not be identifiable from trial data.”

(4) *Where substudies 2 and 3 also registered in a public platform? However, for substudies 2 and 3, I think an informed consent is needed*

Response: the survey (study 2) and qualitative interviews (study 3) were registered as part of the trial registration clinicaltrials.gov (Identifier: NCT05408169, registration date: 07.06.2022). As we state in response to the Editor's comment 2, completion of the postal questionnaire will be taken as an indication of informed consent. It is our experience that sending an informed consent form with a postal questionnaire can confuse participants and they may believe the form is related to another piece of research. The qualitative interview participants will provide informed consent.

(5) *I have doubts on how the sample size is calculated. Usually, sample size is established based on the minimum important difference that is expected to detect between groups in the main outcome (difference in participation rates), given an alpha and beta errors, and considering a proportion of dropouts. Authors just do the inverse thing: based on their sample, they calculate what is the minimum difference in participation that they would be able to detect at a determinate level of significance. I am not sure if this way of explaining sample size calculation is correct. Also, I think there is a mistake in line 3 of page 9: instead of 6.68%, should it be 66.85%?*

Response: Firstly, thank you for noting the mistake which we also noted post submission. We have now corrected the manuscript to '66.85%'. Regarding the sample size, we viewed a 3% absolute increase as a minimum important difference (as this value has been seen before). Given that we were not limited in terms of sample size, we did not calculate the precise number needed to detect this effect but chose a round number of 5000 in each intervention combination. We show that this would have good power to detect 3% differences between the each deadline and the no deadline group (which is the least powerful comparison).

(6) *Data analysis plan: Before conducting any multivariate model, a descriptive analysis and a bivariate analysis should be conducted.*

Response: For clarification, primary and secondary outcome measures will be summarised using descriptive statistics. We have now added the following to the 'Data analysis plan': "Primary and secondary outcome measures will be summarised using descriptive statistics." However, the primary analysis is based on an a priori model specification. Given the factorial design, the analysis of each intervention effect must be adjusted for the other intervention (as would a trial with a stratified randomisation). Since we are using logistic regression for the primary analysis, we will also adjust for factors that we believe will be associated with the primary outcome (age, sex, and socioeconomic deprivation), to reduce bias in the intervention effect estimates.

(7) *For substudies 2 and 3, I would suggest to use the same structure that you have used for the methods section of the RCT: objective, study design, population, sample size, etc..*

Response: Thank you, we have changed the structure as this issue was also raised by Reviewer 2, comment 6.

(8) *Regarding the substudy 2, please provide a definition on who will be the cases, who will be the controls and how will you select these samples. Would you apply any matching strategy between cases and controls?*

Response: We have now clarified in the methods that the following, "The aim is to obtain responses from approximately 1,000 people who have returned the FIT (case) and 1,000 who have not returned (control) the FIT." We will randomly select the subsample while ensuring we have a range of people from deprived and affluent neighbourhoods (based on the Scottish Index of Multiple Deprivation: SIMD), gender and age.

(9) *I would like to suggest to authors to consider the use of electronic questionnaires and post-mail paper questionnaires and let participants choose the option they prefer. This can increase the response rate.*

Response: Thank you for this suggestion. This was considered in designing the research but resource limitations necessitated streamlining the data collection process to a single approach rather than managing both online and offline versions. If we were to undertake a similar study we would certainly consider offering participants a choice if evidence suggests this would increase the response rate.

- (10) *I have some concerns on how the power analysis of substudy 2 is conducted and reported. Sample size calculation is explained as if the main objective of the study was to find the OR of being a case or a control, and I think it makes no sense as authors select who will be cases and who will be controls. The objective of the study is to find differences in the prevalence of some “expositions” (cognitive and behavioural mechanisms) between cases and controls.*

Response: No, we do not select who will be cases or controls – we cannot pick who will return the FIT or not. Of those who return the kit (cases), we randomly select a sample to send questionnaires. The same approach will be used for those who do not return the kit (controls).

- (11) *Please, explain is the questionnaires that will be used to measure acceptability are validated and in a population similar to yours and if they have good measurement properties.*

Response: We have revised the ‘Materials and procedure’ section:

“A brief questionnaire includes items to assess cognitive and behavioural mechanisms, e.g. action and coping planning, that were previously adapted for (but not validated in) the CRC screening context. It also includes a theory-informed acceptability measure that has been subject to a pre-validation method but does not yet have fully established psychometric properties, as well as demographic characteristics. Participants will be asked if they are willing to participate in future research to assist recruitment for qualitative interviews.”

12. *Substudy 3. This is a qualitative study, but authors should explain which method are they going to follow phenomenology? Grounded theory?...*

Response: In this study we aim to explore individuals' experiences (phenomenology). We have revised the ‘Qualitative interviews’ section:

“We will conduct a qualitative interview study, adopting a phenomenological approach to explore participants' experiences of receiving a FIT with and without the interventions, and explore acceptability of the interventions.”

13. *Regarding the sample size for the qualitative study, how do authors know that they would need 40 participants? Although they have estimated a sample size, sampling should continue until data saturation is reached (or it should be stopped is saturation is reached with less people). Recruitment, interviewing and analysis should be done at the same time to detect data saturation.*

Response: The notion of sample size and data saturation within qualitative research remains much debated^[1] but we aim to achieve data saturation whilst considering data adequacy and richness². We have added the following sentence under the ‘Recruitment’ heading:

“The estimated sample size is based on the principles of achieving data saturation and what is pragmatic, yet informative.”

14. *Please, provide a topic guide for the semi-structured interviews in an annex.*

Response: We have included the topic guide in Supplementary file 3.

15. *I would suggest authors to consider the conduction of focus groups (segregated by different profiles) instead of individual interviews. Individual interviews are an appropriate method for gathering data for their objective. However, focus groups allow interaction within participants and this interaction can let the emergence of new information.*

Response: Thank you for this suggestion. We agree that focus groups can generate new information and could be particularly useful in establishing a consensus around intervention acceptability. However, for this study we believe *individual* interviews are more appropriate for our purpose of understanding people's individual experiences of using the interventions. This will involve in-depth exploration of individual thought processes and behavioural responses, which we believe is better explored without influence from group members. This may involve discussion of toilet habits and problems encountered taking stool samples, which may cause embarrassment in a group setting, not be disclosed, and thus influence the data collected.

16. *Please, explain how many people will conduct the qualitative analysis. Would you use any software?*

Response: We have now added the following detail to the manuscript in the ‘Qualitative analysis’ section: “We will use NVivo software to undertake coding, manage and apply a working analytical framework, generate framework matrices and summarise and interpret data. [...] BY will lead the analysis with support from KAR, MK and SM. Interpretation of findings may also involve the wider co-investigator group, including the project Patient & Public Representatives.”

17. Authors should provide what methods they will use to guarantee the rigor of data: will how they will triangulate data?

Response: We have expanded the qualitative analysis section to explain how we will ensure rigour and triangulate data:

"The interviews will be audio-recorded, transcribed verbatim, checked by the interviewer and analysed using the Framework Method. This method facilitates systematic, rigorous and transparent data management, providing an audit trail from raw data to final themes. We will use NVivo software to undertake coding, manage and apply a working analytical framework, generate framework matrices and summarise and interpret data. This method will adopt a combined deductive and inductive approach, guided by the behavioural mechanisms targeted by the interventions (e.g. action and coping planning), as well as constructs within the theoretical framework of acceptability, whilst also allowing other experiences and any unintended intervention effects to be explored. BY will lead the analysis with support from KAR, MK and SM. Interpretation of findings may also involve the wider co-investigator group, including the project Patient & Public Representatives, for improved credibility of findings. In addition, the results from the survey will be used to inform the analysis of the qualitative interviews and vice versa to determine whether synthesising the two data sources can provide unique insights which would not be apparent by analysing the data independently. Calls to the Scottish Bowel Screening Programme helpline relating to the suggested deadline for FIT kit return and planning tool will also be monitored via a call report form including date, time and nature of query as an additional assessment of acceptability."

Reviewer: 4

Dr. Ajay Kumar Khanna, Banaras Hindu University

The authors have planned the study protocol well and have addressed all the mentioned research questions. Following are my comments on the study protocol:

1. *Sample-size: Page 9, line 3: Please check the typographical error*

"increase in uptake rates from 65% to 66.85%. Alternatively, if the intervention increases uptake rates from 60.5% to 68%, the 97.5% confidence interval for the difference in uptake rates will have a width of $\pm 1.06\%$."

Response: Thank you for noting the typographical error which we also noted post submission. We have now corrected the manuscript to '66.85%'.

2. *Will the participants in the intervention group receive some form of reward if they meet the deadlines?*

Response: No, there is no incentive to meet the suggested deadline.

3. *As stated in the manuscript, no participant will be at a disadvantage if they don't meet the deadlines. How will the data for those participants who undertake screening after the deadlines will be part of the data analysis?*

Response: The primary outcome is the proportion of FITs returned correctly completed to be tested by the CRC screening laboratory providing a positive or negative result, within three months of the FIT being sent to a person. However, a secondary analysis will examine whether participants 'met' the deadline.

1

[1] Braun V, Clarke V. To saturate or not to saturate? Questioning data saturation as a useful concept for thematic analysis and sample-size rationales. *Qualitative Research in Sport, Exercise and Health*. 2019; 27:1-6.

[2] Vasileiou K, Barnett J, Thorpe S, Young T. Characterising and justifying sample size sufficiency in interview-based studies: systematic analysis of qualitative health research over a 15-year period. *BMC Medical Research Methodology*. 2018; 18:1, 148.

VERSION 2 – REVIEW

REVIEWER	de Jonge, Lucie Erasmus Medical Center, Public Health
REVIEW RETURNED	13-Apr-2023

GENERAL COMMENTS	Happy with the answers. No further comments.
--

REVIEWER	Goodwin, Belinda Cancer Council Queensland
REVIEW RETURNED	01-Mar-2023

GENERAL COMMENTS	The authors have addressed all my comments. Thank you.
--